# CONSISTENT ITERATIVE DENOISING FOR ROBOT MANIPULATION

## ABSTRACT

Robot manipulation in complex scenarios usually involves multiple successful actions, which requires generative models to estimate the distribution of various successful actions. In recent years, the diffusion model has been widely studied in many robot manipulation tasks. However, the diffusion model experiences inconsistent noise supervision across various action labels and denoising timesteps, which compromises accurate action prediction. On the one hand, CIDM designs new noise supervision to avoid interference between different successful actions, leading to consistent denoising directions. On the other hand, CIDM unifies all denoising timesteps, avoiding inconsistent predictions of the diffusion model over different timesteps. Moreover, we also designed a novel radial loss to make the model focus on denoising results rather than iterative process routes. Our method achieves a new state-of-the-art performance on RLBench with the highest success rate of 82.3% on a multi-view setup and 83.9% on a single-view setup.

## 1 INTRODUCTION

As an important research field of embodied intelligence, robot arm manipulation has a wide range of real-world application scenarios and attracts widespread attention. Robot manipulation mainly involves two steps, acquiring effective scene representation and predicting correct actions. Due to the complexity of action strategies in challenging scenarios, there is an increasing interest in the policy network, which predicts well-performed actions. Additionally, introducing more complex tasks with a diverse set of successful actions puts an extra burden on action predicting (Jia et al., 2024). Traditional regression models as policy networks can only predict a single action, making it difficult to understand scenes with multiple successful actions. Benefiting from the ability to model the distribution of multiple actions, generative models show superior performance in many robotic manipulation tasks.

Among different generative models, the diffusion model achieves leading performance in many visual generation tasks. So far, a series of works inspired by the visual generation, make progress on robotic manipulation using the diffusion model. Diffusion Policy (Chi et al., 2023) finds that diffusion formulation has a strong advantage of robust manipulation and exhibits impressive training stability. Imitating Diffusion (Pearce et al., 2023) discover that diffusion models are suitable for learning from sequential robotic demonstrations. READ (Oba et al., 2024) designs an asymmetric denoising process motivated by Cold Diffusion (Bansal et al., 2024).

However, different from the high sampling density in the visual generation task, robot manipulation has a high data acquisition cost (Cui et al., 2023). It is hard for the diffusion model to learn the accurate probability distribution with inadequate training data. Specifically, the above difficulty of learning an accurate diffusion model for action denoising mainly comes from two aspects:

1) **Difficulty in clarifying an accurate denoising direction.** Since the diffusion model may produce the same noisy action over different successful actions (Ho et al., 2020), the diffusion model will be confused about the denoising directions, leading to inaccurate noise prediction. Especially in the initial denoising stage, the similar initial distributions of successful actions induce severe confusion in the denoising direction. For the sake of clarity, assuming a simple scenario containing two successful actions with equal prior probability, the denoising process of the diffusion model is shown in Figure 1(a). For the noisy action (black point) sampled from the initial noisy distribution, the diffusion model struggles to distinguish whether the denoising is aimed at the blue successful

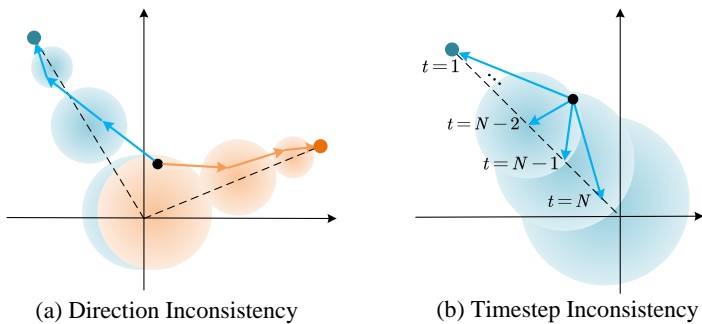

(a) Direction Inconsistency      (b) Timestep Inconsistency

Figure 1: **Difficulties of the diffusion model.** (a) shows the confusion of different denoising directions in the scenario with two successful actions, and (b) shows the inconsistent noise supervision over different timesteps.

action or the red successful action. As a result, the diffusion model faces the mutual interference of multiple successful actions, leading to inaccurate denoised actions.

2) **Difficulty in learning a time-varying denoising model.** In the diffusion model, the time-varying noise addition process forms a series of time-varying distributions of noisy actions. In order to generate accurate actions through iterative denoising, the diffusion model needs to learn the denoising ability over all timesteps. As shown in Figure 1(b), for the specific noisy action (black point), the noises supervision signals (blue arrows) are inconsistent at different timesteps. The temporal inconsistency of the diffusion model increases the difficulty of iterative denoising, which reduces the accuracy of denoised actions from a practical perspective.

To address the above difficulties, we hope to build a more consistent denoising process, through which all noisy actions in the action space could be correctly denoised. As low-dimension action space is easy to cover by action samples, different from image generation space (Section A.1 in Appendix), robot manipulation prefers to sample initial actions over the entire action space, rather than restricted to the standard Gaussian distribution. For example, 3D Diffuser Actor (Ke et al., 2024) already samples the initial noisy action from a Gaussian distribution with a non-zero mean and achieves better training results. Based on the flexibility of initial action distributions, it becomes feasible to design a more consistent denoising process.

In this paper, we propose a novel CIDM to predict more accurate denoised actions in multiple tasks. For a specific scenario, CIDM establishes a denoising field in the action space, which gives noise prediction for arbitrary noisy action. On the one hand, we design a more consistent denoising field in action space, which supplies noise supervision with clear directions during training. On the other hand, we train the CIDM in a time-invariant fashion to ensure the consistency of noise predictions over different timesteps, avoiding the difficulty of learning a time-varying representation. Additionally, We also propose a novel radial loss to pay more attention to the action samples with little noise, which enables the iterative denoising process to converge more accurately. Our contributions can be summarized as follows:

- Through theoretical analysis, we identify the shortcomings of the diffusion model in the action space and gain inspiration for iterative denoising.

- We design a consistent iterative denoising model for robot manipulation, which builds the denoising field with clear denoising directions and temporal consistency.

- We propose a new radial loss function to emphasize action samples with small noises and achieve a more robust iterative denoising process.

- We evaluate our method on RLBench tasks, it achieves state-of-the-art performance with the highest average success rate. We also verify the effectiveness of our components through ablation experiments.

## 2 RELATED WORK

**Diffusion model.** Through the iterative denoising process, early diffusion models (Ho et al., 2020; Song et al., 2021a) enable diverse and high-quality visual generation. Since the diffusion process could be modeled as a stochastic differential equation (Song et al., 2021b), the continuous diffusion models (Dockhorn et al., 2022; Jolicoeur-Martineau et al., 2021) achieve more efficient generation with fewer steps. The widely-regarded stable diffusion adopts a latent space (Rombach et al., 2022; Vahdat et al., 2021) to achieve computational efficiency, while the lower bound on the dimension of the latent space is still limited by the need to decode images. In the last few years, more than speeding up the denoising process, recent work has also provided a more in-depth analysis of diffusion models. Cold Diffusion (Bansal et al., 2024) designs a more robust iteration to revert arbitrary degradation. Inversion by Direct Iteration (Delbracio & Milanfar, 2023) pursues a simpler form to get rid of the limitations of traditional diffusion. Some recent works Lin et al. (2024); Zhang et al. (2024) have noticed and attempted to address the subtle differences in sampling distributions between training and inference, which were previously ignored. Research on a few samples (Wu et al., 2024b) is done through fine-tuning rather than complete retraining, which also reflects the dependence of diffusion models on sufficient training data.

**Diffusion model in robotic manipulation.** In recent years, a series of works have verified the potential of diffusion models in robot manipulation. Different from using diffusion models to generate more visual scene information (Wu et al., 2024a), the potential of diffusion models to predict actions has also been explored. Diffusion Policy (Chi et al., 2023) successfully models the probability of trajectory sequences in different tasks. 3D Diffusion Policy (Ze et al., 2024) incorporates the power of 3D visual representations into conditional diffusion models. DNActor (Yan et al., 2024) distill 2D semantic features from foundation models, such as Stable Diffusion (Rombach et al., 2022) and state representation on NeRF (Driess et al., 2022), to a 3D space in its pretrain phase. With the continuous improvement of diffusion models in the field of visual generation, works are designing new diffusion paradigms in robotic manipulation. Hierarchical Diffusion Policy (Ma et al., 2024) adds a new robot kinematic constraint on the diffusion models. READ (Oba et al., 2024) preserves the kinematic feasibility of the generated action via forward diffusion in a low-dimensional latent space, while using cold diffusion to achieve high-resolution action via back diffusion in the original task space. These methods inspire us to enhance the diffusion model for generating actions that align with the characteristics of robot manipulation.

## 3 ITERATIVE CONSISTENT DENOISING MODEL

To provide a clearer explanation of the background and our method, this section is organized as follows: (1) We start by introducing notations and analyzing existing difficulties in the Preliminaries. (2) Then we introduce the Overview of the consistent iterative denoising model (CIDM). (3) Finally, we analyze the rationality of the two main components of CIDM: Consistent Denoising Field and Radial Loss Function.

### 3.1 PRELIMINARIES

**Robot manipulation**. The key-frame robot manipulation is described by a sparse sequence of the robot trajectory and corresponding scene information $x$, which contains multi-view RGB-D images, the text instruction, and the current robot state. As parallel gripper robot arms interact with the environment through the end effector, we use end-effort posture $y$ to guide the action of the robot arm, which includes the translation, the rotation, and the binary opening state of the gripper. Due to the opening state containing little location information, only the translations and the rotations are input as noisy actions. In the successful demonstrations for training, each scene $x$ corresponds to a label action $\hat{y}$, which is one of the successful actions $\{\hat{y}^i\}_{i=1}^k$.

**Itervatively denoising process.** The diffusion model is a typical iterative denoising method, which has been widely used in robot manipulation. During the training of the diffusion model with timestep $t \in \{1, 2, ..., N\}$, the noise addition is as the following formula:

$$y_t = \overline{\alpha}_t \hat{y} + \sqrt{1 - \overline{\alpha}_t^2}\varepsilon \,, \ \varepsilon \sim \mathcal{N}(0, I), \tag{1}$$

where the action noise $\varepsilon$ is predicted by the diffusion model $\varepsilon_\theta(x, y, t)$ with learnable parameters $\theta$. During inference, the diffusion model randomly samples a noisy action $y_N$. After denoising for $N$ steps, the diffusion model produces the denoised action $y_0$.

**Difficulties of diffusion.** Since successful actions $\{\hat{y}^i\}_{i=1}^k$ usually have the same prior probability, the noisy distribution can be expressed as:

$$p_t(y_t) = \sum_{i=1}^k p_t(y_t|\hat{y}^i)p(\hat{y}^i) = \frac{1}{k} \sum_{i=1}^k \mathcal{N}(y_t; \overline{\alpha}_t \hat{y}^i, \sqrt{1 - \overline{\alpha}_t^2} \varepsilon_t). \tag{2}$$

When the scene information $x$ and denoising timestep $t$ are determined, the optimization of the diffusion model $\varepsilon_\theta(x, y, t)$ is as follows (Song & Ermon, 2019):

$$\theta = \arg \min_\theta \mathbb{E}_{p_t(y_t|\hat{y})p(\hat{y})}[\lambda(t)\|\nabla_{y_t} \log p_t(y_t|\hat{y}) - \varepsilon_\theta(x, y_t, t)\|_2^2]. \tag{3}$$

After eliminating the effects of specific successful action $\hat{y}$, the optimization process of the diffusion model can also be expressed as follows:

$$\theta = \arg \min_\theta \mathbb{E}_{p_t(y_t)}[\lambda(t)\|\nabla_{y_t} \log p_t(y_t) - \varepsilon_\theta(x, y_t, t)\|_2^2]. \tag{4}$$

Therefore, $\varepsilon_\theta(x, y_t, t)$ learns to represent the score function $\nabla_{y_t} \log p_t(y_t)$, which is independent of specific successful action $\hat{y}$. For different noisy actions $y_t$, the diffusion model constructs a denoising field in the action space, which is ideally equivalent to $\nabla_{y_t} \log p_t(y_t)$.

The first problem is that the score function $\nabla_{y_t} \log p_t(y_t)$ is biased as a denoising field. The $t$-th denoising is towards the actions with zero noise $\nabla_{y_t} \log p_t(y_t) = 0$, which has the local maximum probability in distribution $p_t(y)$ and satisfies the following condition:

$$\frac{\mathrm{d}p_t(y_t)}{\mathrm{d}y_t} = p_t(y_t) \nabla_{y_t} \log p_t(y_t) = 0. \tag{5}$$

However, since $p_t(y_t)$ is a mixed Gaussian distribution, the action $\overline{\alpha}_t \hat{y}^i$ are not the local maximum probability in $p_t(y_t)$:

$$\begin{aligned}
\left. \frac{\mathrm{d}p_t(y_t)}{\mathrm{d}y_t} \right|_{y_t = \overline{\alpha}_t \hat{y}^j} &= \left[ \frac{1}{k} \sum_{i=1}^k \mathcal{N}(y_t; \overline{\alpha}_t \hat{y}^i, \sqrt{1 - \overline{\alpha}_t^2} \varepsilon_t) \right]'_{y_t = \overline{\alpha}_t \hat{y}^j} \\
&= \frac{1}{k} \sum_{\substack{i=1 \\ i \neq j}}^k \left[ \mathcal{N}(y_t; \overline{\alpha}_t \hat{y}^i, \sqrt{1 - \overline{\alpha}_t^2} \varepsilon_t) \right]'_{y_t = \overline{\alpha}_t \hat{y}^j} \neq 0.
\end{aligned} \tag{6}$$

Furthermore, interference between successful actions will be more severe at the initial denoising stage, leading to a combination of all successful actions $\frac{1}{k} \sum_{i=1}^k \overline{\alpha}_N \hat{y}^i$ (Section A.2 in Appendix).

Another problem comes from the time-varying characteristic of the diffusion model. When a fixed scenario $x$ and successful action $\hat{y}$ are selected, the ideal noise prediction for the same noisy action $y$ changes over timesteps $t$:

$$\varepsilon_\theta(x, y, t) = \frac{y - \overline{\alpha}_t \hat{y}}{\sqrt{1 - \overline{\alpha}_t^2}}. \tag{7}$$

This burden of simultaneously modeling probability distributions over all timesteps affects the accuracy of the diffusion model.

## 3.2 OVERVIEW

In text-guided robotic manipulation, the robot needs to interact with the environment according to the text instruction. Our framework consists of a multi-modal encoder and a time-invariant denoising network to predict noise in an iterative process.

We adopt the CLIP image encoder and text encoder to extract features of visual observations and text instructions respectively. The scene features are obtained through the multi-modal encoder with pretrained parameters as follows:

$$F_x = \text{Encoder}(x), \tag{8}$$

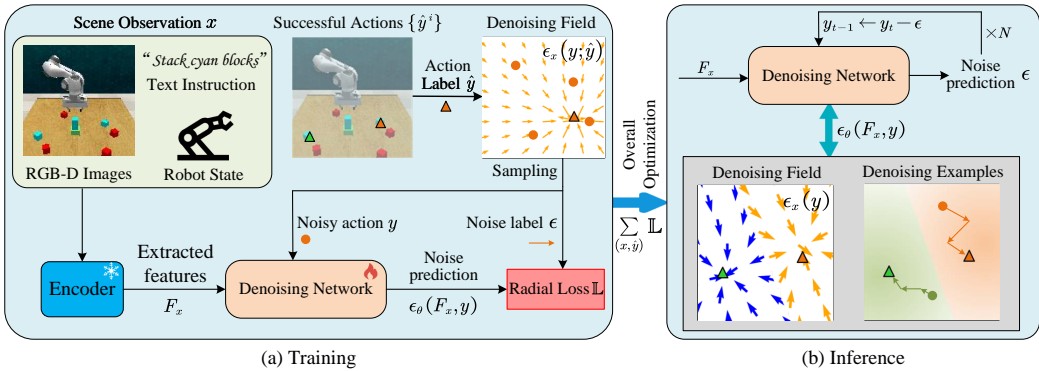

(a) Training

(b) Inference

Figure 2: **The pipeline of CIDM.** CIDM builds a time-invariant denoising field $\epsilon_x(y; \hat{y})$ for training. After training on multiple $(x, \hat{y})$ pairs, the denoising network learns to construct a $\hat{y}$-independent denoising field $\epsilon_x(y)$ for inference.

where $x$ denotes scene information. Then, the multi-modal feature $F_x$ together with a noisy action $y$ are fed into the time-invariant denoising network $\epsilon_\theta$ to predict the action noise $\epsilon$:

$$\epsilon = \epsilon_\theta(F_x, y), \tag{9}$$

where $\theta$ denotes the learnable parameters.

**Training.** As shown in Figure 2(a), we design a time-invariant denoising field $\epsilon_x(y; \hat{y})$, which is conditioned on successful action $\hat{y}$ in the scene $x$. For noisy action $y$, we sample noisy supervisions from the denoised field $\epsilon_x(y; \hat{y})$. The input of the denoising network $\epsilon_\theta$ contains the encoded features $F_x$ and the noisy action $y$. For specific training data pair $(x, \hat{y})$, the denoising network $\epsilon_\theta(F_x, y)$ are optimized towards $\epsilon_x(y; \hat{y})$. Through the guidance of overall loss on all training pairs, the denoising network learns to represent different scenes $x$ as the following denoising field:

$$\epsilon_\theta(F_x, y) \to \epsilon_x(y), \tag{10}$$

where the denoising field $\epsilon_x(y)$ is independent with specific successful action $\hat{y}$ and could achieve a correct denoising.

**Inference.** The iterative denoising process of CIDM is shown in Figure 2(b), where the scene encoding is omitted. First, we randomly sample the initial action $y_N$ from the action space. Then, we iteratively denoise the action through the denoising network as follows:

$$y_{t-1} = y_t - \epsilon_\theta(F_x, y_t), \ t \in \{1, 2, ..., N\}, \tag{11}$$

where $\epsilon_\theta(F_x, y_t)$ learns to represent the denoising field $\epsilon_x(y)$. After iterative denoising for $N$ steps, our CIDM gains accurate action prediction $y_0$.

### 3.3 Consistent Denoising Field

As CIDM relies on the denoising network $\epsilon_\theta(F_x, y)$ for iterative denoising, it is very important to learn from a reasonable denoising field $\epsilon_x(y)$. In an ideal time-invariant denoising field, arbitrary actions $y$ in the action space can reach a successful action within finite steps of denoising. Since iterative denoising results in different successful actions, we divide the action space into distinct regions, each region corresponding to a specific successful action. The reasonable denoising field $\epsilon_x(y)$ as described above complies with two conditions:

(1) Since the reasonable denoising field always makes noisy action closer to its target successful action, there must be a neighborhood of $\hat{y}$ where actions reach $\hat{y}$ through a single-step denoising, expressed as follows:

$$\exists c<0, \forall \|y - \hat{y}\|_2<c, \ \epsilon_x(y) = y - \hat{y}. \tag{12}$$

(2) On the boundaries that separate different regions in the action space, the noise prediction cannot point to either side. To achieve the best symmetry, we set $\epsilon_x(y) = 0$ with $y$ on the boundaries.

Actually, the noise supervision of the denoising network is $\epsilon_x(y; \hat{y})$ during training. To finally learn a reasonable denoising field $\epsilon_x(y)$, we design a consistent $\epsilon_x(y; \hat{y})$. In the scenario with a single successful action, $\epsilon_x(y; \hat{y})$ equals to $\epsilon_x(y)$. Therefore, the denoising field during training has a similar requirement to Equ. (12):

$$\exists\, c<0, \forall\, \|y - \hat{y}\|_2 < c,\ \epsilon_x(y; \hat{y}) = y - \hat{y}. \tag{13}$$

In the scenario with multiple successful actions, $\epsilon_x(y; \hat{y}^j)$ is supposed to have a small difference from $\epsilon_x(y)$ in regions corresponding to $\{\hat{y}^i\}_{i \neq j}$. To satisfy $\epsilon_x(y) = 0$ on the boundaries, a simple idea is to have limited $\|\epsilon_x(y; \hat{y}^j)\|_2$ as $y$ moves away from $\hat{y}^j$. Considering the above requirements, we design a new denoising field during training as follows:

$$\epsilon_x(y; \hat{y}) = \left\{ \begin{array}{ll} y - \hat{y} & \text{for } \|y - \hat{y}\|_2 < c\,, \\ c(y - \hat{y})/\|y - \hat{y}\|_2 & \text{for } \|y - \hat{y}\|_2 \geq c\,, \end{array} \right. \tag{14}$$

where hyperparameter $c$ is smaller than the distance between two successful actions.

By training on all $(x, \hat{y})$ pairs, the denoising network $\epsilon_\theta(F_x, y)$ learns to predict noise without dependence on specific $\hat{y}$. Unlike the diffusion model, which converges to $\overline{\alpha}_t \hat{y}$ at timestep $t$, our CIDM converges to $\hat{y}$ consistently over all timesteps. Due to the unification of our denoising fields over different timesteps, the denoised field $\epsilon_x(y)$ could be more accurately predicted by the denoising network $\epsilon_\theta(F_x, y)$. Consequently, we achieve accurate iterative denoising during inference based on better noise prediction.

### 3.4 RADIAL LOSS FUNCTION

As the design of the $\epsilon_x(y; \hat{y})$ during training is only necessary but not sufficient to learn a reasonable $\epsilon_x(y)$, we design the radial loss function $\mathbb{L}$ to optimize the denoising network $\epsilon_\theta(F_x, y)$. Essentially, the optimization of the denoising network on all training data is as follows:

$$\theta = \arg\min_\theta \mathbb{E}_{p(y|\hat{y})p(x,\hat{y})} \mathbb{L}\big(\epsilon_\theta(F_x, y), \epsilon_x(y; \hat{y})\big). \tag{15}$$

Assuming that the denoising network has sufficient fitting ability, $\epsilon_\theta(F_x, y)$ is supposed to represent the target denoising field $\epsilon_x(y)$ expressed as follows:

$$\epsilon_x(y) = \arg\min_\epsilon \mathbb{E}_{p(y|\hat{y})p(x,\hat{y})} \mathbb{L}\big(\epsilon, \epsilon_x(y; \hat{y})\big). \tag{16}$$

Our radial loss function $\mathbb{L}$ should make $\epsilon_x(y)$ close to $\epsilon_x(y; \hat{y}^j)$ when $y$ is close to $\hat{y}^j$. In addition, a small noise prediction error is acceptable for actions requiring multi-step denoising. Therefore, when noisy action $y$ gets closer to successful actions $\{\hat{y}^i\}_{i=1}^k$, the loss function should pay more attention to it. Based on the above perceptions, we designed a radial loss function as follows:

$$\mathbb{L}(y, \hat{y}) = \delta(\|y - \hat{y}\|_2)\, L_1\big(\epsilon_\theta(F_x, y), \epsilon_x(y; \hat{y})\big), \tag{17}$$

where $L_1$ denotes the $L_1$ loss function and $\delta(\cdot)$ denotes the radial weight as follows:

$$\delta(r) = \min(1/\sqrt{r}, 10). \tag{18}$$

The $\delta(r)$ ensures that small radial distance $r = \|y - \hat{y}\|_2$ corresponds to a large weight in the overall loss. We set an upper bound of 10 for $\delta$ to avoid excessive loss, which leads to unstable training. Compared with $L_2$ loss, the gradient of $L_1$ loss does not increase with a larger prediction error. The $L_1$ loss makes the target denoising field $\epsilon_x(y)$ focus on specific successful action corresponding to $y$, instead of being affected by all successful actions.

The target denoised field $\epsilon_x(y)$ obtains good properties through the radial loss function. On the one hand, $\epsilon_x(y)$ can also accurately denoise in a single step when $y$ is close to a successful action $\hat{y}$ (Section A.3 in Appendix). On the other hand, $\epsilon_x(y)$ enables the correct denoising of noisy actions that are far from successful actions. Therefore, our radial loss helps to learn an accurate and consistent denoising process.

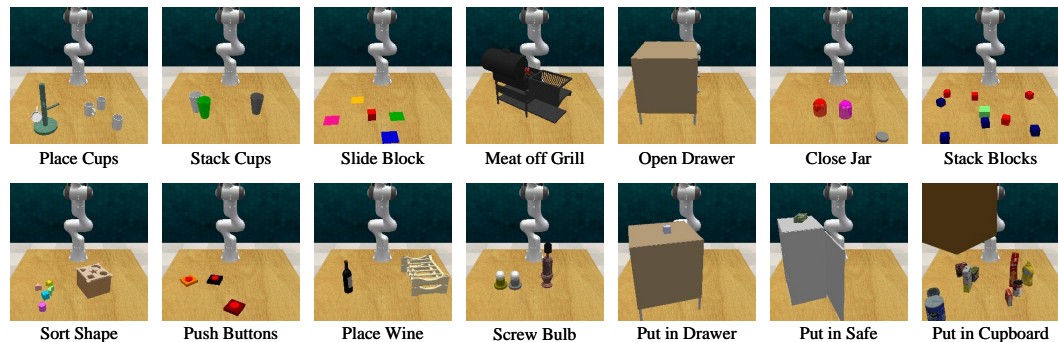

Figure 3: **List of 14 tasks.** These challenging tasks are highly representative.

## 4 EXPERIMENTS

### 4.1 EXPERIMENTAL SETUP

**Dataset and simulation.** We evaluate our CIDM on a multi-task manipulation benchmark developed in RLBench (James et al., 2020). We use 18 distinct tasks, each comprising 150 demonstrations, with 14 of them illustrated in Figure 3. Each task includes various text instructions, which feature between 2 to 60 variations. These variations consist of several types, such as variations in position and color. The demonstrations are collected in a simulation environment built by CoppeliaSim (Rohmer et al., 2013). The $256 \times 256$ RGB-D images in demonstrations are captured by four noiseless cameras positioned at the front, left shoulder, right shoulder, and wrist of the robot. In addition, we verify our performance in the simulation environment.

**Training and evaluation details.** Our CIDM is trained on 4 NVIDIA 3090Ti 10GB GPUs for 60K steps with a cosine learning rate decay schedule. We adopt a batch size of 32 and initialize the learning rate to $10^{-4}$. Among the 150 pre-generated demonstrations, 100 are used for training, 25 for validation, and 25 for testing. During training, we prefer to sample points close to successful actions. We evaluate CIDM in both multi-view and single-view settings. As the action planner in the simulation environment has a certain degree of randomness, we evaluate each task four times and take the average success probability as the performance metric.

**Baselines.** In text-guided robot manipulation, we compare CIDM with the existing baselines, which have made significant advancements and achieved excellent performance. The following work has improved scene representation methods in robotic manipulation: PolarNet (Chen et al., 2023), Hive-Former (Guhur et al., 2023), PerAct (Shridhar et al., 2023), Act3D (Gervet et al., 2023), RVT (Goyal et al., 2023). Additionally, RVT2 (Goyal et al., 2024) using action-value graphs and 3D Diffuser Actor Ke et al. (2024) using the diffusion model make progress in modeling multiple successful actions.

### 4.2 MAIN RESULTS

Following the setting of PerAct (Shridhar et al., 2023), we conduct experiments on 18 tasks with 4 camera views. As shown in Table 1, our CIDM achieves the sota performance among existing methods, boasting the highest average success rate of 82.3% and the best average ranking of 2.0 across all tasks. Specifically, our CIDM secures the best performance on 7 tasks and the suboptimal performance on 7 tasks. Moreover, compared with the diffusion-based model 3D Diffuser Actor, CIDM improves performance on tasks with multiple successful actions. Performance degradation on some tasks is caused by multi-task training, where success rates fluctuate to some extent.

Following the setting of GNFactor (Ze et al., 2023), we also conduct experiments on 10 tasks (a subset of the 18 tasks) with a single view. As shown in Table 2, our CIDM achieves the highest average success rate of 83.9%. Significant performance improvements are achieved by CIDM on 5 tasks, including *meat off grill, turn tap, put in drawer, push buttons, stack blocks*. In particular, we significantly improved performance on the most difficult task *stack blocks*.

Table 1: **Evaluation on RLBench with multiple camera views**. Our approach achieved the highest average task success rate. Black bold fonts indicate the best performance and underline indicate suboptimal performance for each column.

| Models | Avg. Success. | Avg. Rank. | Place Cups | Stack Cups | Sort Shape | Push Buttons | Stack Blocks | Put in Cupboard | Slide Block | Meat off Grill |
|---|---|---|---|---|---|---|---|---|---|---|
| PolarNet | 46.4 | 6.4 | 0 | 8 | 12 | 96 | 4 | 12 | 56 | **100** |
| PerAct | 49.4 | 6.2 | 2.4 | 2.4 | 16.8 | 92.8 | 26.4 | 28 | 74 | 70.4 |
| HiveFormer | 45 | 6.6 | 0 | 0 | 8 | 84 | 8 | 68 | 64 | **100** |
| Act3D | 65 | 4.4 | 3 | 9 | 8 | 99 | 12 | 51 | 93 | 94 |
| RVT | 62.9 | 4.6 | 4 | 26.4 | 36 | **100** | 28.8 | 49.6 | 81.6 | 88 |
| RVT2 | 81.4 | 2.4 | **38** | **69** | 35 | **100** | **80** | 66 | 92 | 99 |
| 3D Diffuser Actor | 81.3 | 2.4 | 24 | 47.2 | 44 | 98.4 | 68.3 | **85.6** | 97.6 | 96.8 |
| **CIDM (Ours)** | **82.3** | **2.0** | 32 | 53 | **48** | 98 | 69 | 76 | **100** | 98 |

| Models | Open Drawer | Close Jar | Place Wine | Screw Bulb | Put in Drawer | Put in Safe | Drag Stick | Insert Peg | Sweep to Dustpan | Turn Tap |
|---|---|---|---|---|---|---|---|---|---|---|
| PolarNet | 84 | 36 | 40 | 44 | 32 | 84 | 92 | 4 | 52 | 80 |
| PerAct | 88 | 55.2 | 44.8 | 17.6 | 51.2 | 84 | 89.6 | 5.6 | 52 | 88 |
| HiveFormer | 52 | 52 | 80 | 8 | 68 | 76 | 76 | 0 | 28 | 80 |
| Act3D | 93 | 92 | 80 | 47 | 90 | 95 | 92 | 27 | 92 | 94 |
| RVT | 71.2 | 52 | 91 | 48 | 88 | 91.2 | 99.8 | 11.2 | 72 | 93.6 |
| RVT2 | 74 | **100** | 95 | **88** | **96** | 96 | 99 | 40 | **100** | 99 |
| 3D Diffuser Actor | 89.6 | 96 | 93.6 | 82.4 | **96** | 97.6 | **100** | **65.6** | 84 | **99.2** |
| **CIDM (Ours)** | **93** | 96 | **96** | 80 | **96** | **100** | **100** | 54 | 97 | 97 |

Table 2: **Evaluation on RLBench with single camera view.** We report success rates on 10 RL-Bench with only the *front* camera view.

| Models | Avg. Success. | close jar | open drawer | sweep to dustpan | meat off grill | turn tap | slide block | put in drawer | drag stick | push buttons | stack blocks |
|---|---|---|---|---|---|---|---|---|---|---|---|
| GNFactor | 31.7 | 25.3 | 76.0 | 28.0 | 57.3 | 50.7 | 20.0 | 0.0 | 37.3 | 18.7 | 4.0 |
| Act3D | 65.3 | 52.0 | 84.0 | 80 | 66.7 | 64.0 | 100.0 | 54.7 | 86.7 | 64.0 | 0.0 |
| 3D Diffuser Actor | 78.4 | **82.7** | **89.3** | 94.7 | 88.0 | 80.0 | **92.0** | 77.3 | 98.7 | 69.3 | 12.0 |
| **CIDM (Ours)** | **83.9** | 78 | 88 | **98** | 92 | 85 | 90 | 91 | 100 | 96 | 21 |

To illustrate the advantages of our method more vividly, we visualize the iterative denoising process in a specific scenario. As shown in Figure 4, the robot arm was ordered to *stack two red blocks*, and the coordinates of red blocks are marked with red triangles in the desktop coordinate system. We sample initial actions in the desktop flat and visualize their positions during iterative denoising. By comparing the coordinates of the denoised actions and the red triangles, we divide the noisy actions into correct denoising (blue points) and incorrect denoising (gray points). Although many initial actions are incorrectly denoised through the diffusion-based model (Ke et al., 2024), our CIDM exhibits greater robustness to different initial actions, owing to the spatial and temporal consistency of the designed denoising field. The robot actions are visualized in Appendix A.4.

## 4.3 ABLATIONS AND ANALYSES

In this section, We conduct a series of ablation studies to assess the effectiveness of the different components in our proposed method. Based on the results of the ablation experiments, we provide a brief analysis of the underlying reasons.

**Ablation on sampling strategy.** Similar to the diffusion model, we use a central sampling manner to get noisy actions during training, where noisy actions $y$ close to successful action have a higher probability of being sampled. In row 2, we utilize the uniform distribution in action space to sample noisy actions. Since central sampling focuses more on successful actions, it gains an improvement of 7.3% success rate, emphasizing the importance of small noise actions.

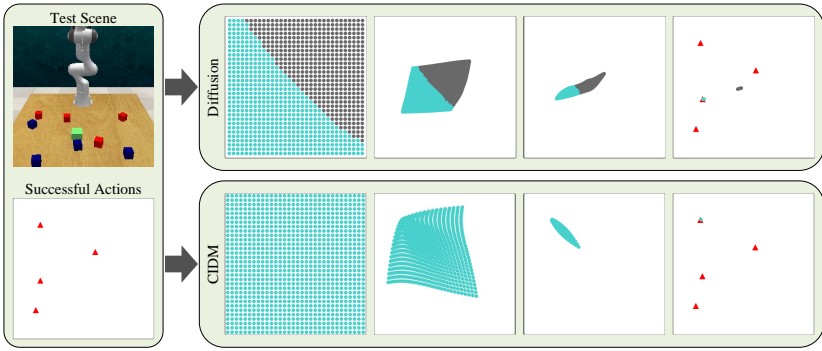

Figure 4: **Visualization of the task to stack two red blocks.** Red triangles denote red blocks. Blue points are denoised into successful actions and gray points are denoised into wrong actions.

Table 3: Some Ablations on RLBench

| Row ID | Central Sampling | Consistent Denoising Field | Radial Loss | Avg. Success. | Avg. Success. diff. wrt. base |
|--------|-----------------|----------------------------|-------------|---------------|-------------------------------|
| 1 | ✓ | ✓ | ✓ | 82.3 | 0 |
| 2 | ✗ | ✓ | ✓ | 75.0 | -7.3 |
| 3 | ✓ | ✗ | ✓ | 79.5 | -2.8 |
| 4 | ✓ | ✓ | ✗ | 79.3 | -3.0 |

**Ablation on denoising field.** To variation the efficiency of our denoising field during training, we use the denoising field $\varepsilon_x(y; \hat{y}) = y - \hat{y}$ of the diffusion model in row 3, comparing with the consistent denoising field $\epsilon_x(y; \hat{y})$ in row 1. From row 1 and row 3, our proposed consistent denoising field achieves a success rate improvement of 2.8% in multiple tasks.

**Ablation on loss function.** As shown in Table 3, we conduct the ablation experiment to verify the effect of the radial loss function. In row 1, the radial loss sets higher weights to noisy actions with smaller noise. In row 4, we use the $L_2$ loss function, the same as the diffusion model. From row 1 and row 4, the radial loss obtains a success rate improvement of 3.0% by focusing more on the neighborhood of successful actions.

Table 4: Ablation on temporal consistency

| Time Coefficient $\overline{\alpha}_N$ | 0.01 | 0.5 | 1 |
|----------------------------------------|------|-----|---|
| Avg. Success Rate | 74.8 | 80.8 | **82.0** |

**Ablation on time variability.** To analyze the importance of temporal consistency, we used a time-variable denoising field $\epsilon_x(y; \overline{\alpha}_t \hat{y})$ during training with the time coefficient $\alpha_t$. Following the time-varying denoising process of the diffusion method, time coefficient $\overline{\alpha}_t$ decreases from $\overline{\alpha}_0 = 1^-$ to $\overline{\alpha}_N$. With the same steps $N = 100$, the smaller $\overline{\alpha}_N$ corresponds to the larger time variation, which is more difficult for the denoising network to represent. As shown in Table 4, the effectiveness of temporal consistency with $\overline{\alpha}_N = 1$ has been verified by its leading performance.

## 5 CONCLUSION

In this paper, we propose the consistent iterative denoising model (CIDM) for text-guided robot manipulation. We build a more consistent denoising field than the diffusion model, by designing noise supervision and unifying the timesteps during training. Moreover, utilizing the radial loss, CIDM avoids interference from other successful actions and obtains accurate denoised actions. On diverse simulated robot manipulation tasks, CIDM achieves state-of-the-art performance in both multi-view and single-view settings. Ablation studies are conducted on various components within CIDM, providing further clarity on their efficiency.

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

# A APPENDIX

## A.1 GAP BETWEEN ROBOT MANIPULATION AND IMAGE GENERATION

In the iterative denoising process for robot manipulation and image generation, the denoising network needs to denoise the samples in an Euclidean space. Due to the powerful representation ability of deep networks, we need a large number of discrete samples during training. If the Euclidean space is discretized into a point set with a sufficiently small distance $\nu$, the continuity of the network ensures that represent on the discrete point set $\{x\}$ could be approximately interpolated into the continuous space:

$$f(x + \gamma\nu) = (1 - \gamma)f(x) + \gamma f(x + \nu) + o(\nu), \tag{19}$$

where $x, x + \nu \in \{x\}$, and $o(\nu)$ denotes the higher-order infinitesimals and $0 < \gamma < 1$. As $\nu$ takes a sufficiently small value, $o(\nu)$ becomes negligible. Since the value range of images and robot actions is bounded, their space can be normalized as follows:

$$\mathbf{x} = [x^{(1)}, x^{(2)}, ..., x^{(n)}] \in \mathbb{R}^n, \|x^{(i)}\| \leq 1. \tag{20}$$

By gridding we can cover the sampling space with as few points as possible, the approximate number of points is estimated as $\left(\frac{2}{\nu}\right)^n$. Since common image spaces correspond to $n > 10^4$, the denoising network can't fit the denoising field at so many points simultaneously. Thanks to the low-dimension action space with $n = 9$, we can train the denoising network on the entire action space.

## A.2 DENOISING ON INITIAL NOISY ACTIONS

During the training of the diffusion model, it is necessary to sample successful actions $\hat{y}$, denoising timesteps $t$, and noise $\varepsilon$. The complete loss function in terms of conditional probability is expressed as follows:

$$loss = \mathbb{E}_{t,p_t(y|\hat{y}),p(x,\hat{y})}[\lambda(t)\|\nabla_x \log p_t(y|\hat{y}) - \varepsilon_\theta(x, y, t)\|_2^2]. \tag{21}$$

When the network $\varepsilon_\theta(y, t)$ can well fit the score function at all times, it can be considered that there is no significant conflict between the optimization of model parameters at different times. So that we can break down the parameter optimization at a specific timestep $t$ and scene $x$:

$$loss_{(t,x)} = \lambda(t)\mathbb{E}_{p_t(y|\hat{y}),p_x(\hat{y})}\|\nabla_x \log p_t(y|\hat{y}) - \varepsilon_\theta(x, y, t)\|_2^2. \tag{22}$$

According to the DDPM noise addition in Equ. (1)and the discrete prior distribution of successful actions $p_x(\hat{y}^i) = \frac{1}{k}, i \in \{1, 2, ..., k\}$, we can further obtain :

$$loss_{(t,x)} = \frac{\lambda(t)}{k}\sum_{i=1}^{k}\int_y p_t(y|\hat{y}^i)\|\nabla_x \log p_t(y|\hat{y}^i) - \varepsilon_\theta(x, y, t)\|_2^2 dy \tag{23}$$

In actual training, we will discretely sample noisy action $\{y_j\}, j \in \{1, 2, ..., M\}$ on distribution $p_t(y|\hat{y}^i)$. In particular, when $t \to N$, all conditional distributions $p_t(y|\hat{y}^i)$ are approximately the same, leading to following formula with $t = N$:

$$loss_{(N,x)} = \frac{\lambda(N)}{k}\sum_{i=1}^{k}\sum_{j=1}^{M}\|\nabla_x \log p_N(y_j|\hat{y}^i) - \varepsilon_\theta(x, y_j, N)\|_2^2 \tag{24}$$

$$= \frac{\lambda(N)}{k}\sum_{j=1}^{M}(\sum_{i=1}^{k}\|\nabla_x \log p_N(y_j|\hat{y}^i) - \varepsilon_\theta(x, y_j, N)\|_2^2). \tag{25}$$

Assuming the model fitting ability is strong enough, we can get the following formula:

$$\text{if } \hat{\theta} = \arg\min_\theta loss_{(N,x)}, \tag{26}$$

$$\text{s.t. } \varepsilon_{\hat{\theta}}(x, y_j, N) = \arg\min_\varepsilon \sum_{i=1}^{k}\|\nabla_x \log p_N(y_j|\hat{y}^i) - \varepsilon\|_2^2. \tag{27}$$

The result of minimizing the loss function of the model is shown below:

$$\varepsilon_{\hat{\theta}}(x, y_j, N) = \frac{1}{k}\sum_{i=1}^{k}\nabla_x \log p_N(y_j|\hat{y}^i). \tag{28}$$

## A.3 FURTHER ANALYSIS OF RADIAL LOSS

Combine the Equ. (16) and Equ. (17), we can get the ideal target denoising field $\epsilon_x(y)$ as follows:

$$
\begin{aligned}
\epsilon_x(y) &= \arg\min_\epsilon \mathbb{E}_{p(y|\hat{y})p(x,\hat{y})}\big[\delta(\|y-\hat{y}\|_2)L_1\big(\epsilon,\epsilon_x(y;\hat{y})\big)\big] \\
&= \arg\min_\epsilon \mathbb{E}_{p(y|\hat{y})p(x)}\big[p(\hat{y}|x)\delta(\|y-\hat{y}\|_2)\,L_1\big(\epsilon,\epsilon_x(y;\hat{y})\big)\big].
\end{aligned}
\tag{29}
$$

When scene information $x$ and noisy action $y$ are determined, $\epsilon_x(y)$ could be simplified with $p(\hat{y}|x) = \frac{1}{k}$ as follows:

$$
\epsilon_x(y) = \arg\min_\epsilon \frac{1}{k}\sum_{i=1}^{k}\delta(\|y-\hat{y}^i\|_2)\,L_1\big(\epsilon,\epsilon_x(y;\hat{y}^i)\big)
\tag{30}
$$

For arbitrary successful action $\hat{y}^j \in \{\hat{y}^i\}_{i=1}^k$, $\delta(\|y-\hat{y}^j\|_2)$ increases when $y$ gets closer to $\hat{y}^j$. Considering that the number of successful actions $k < max(\delta) = 10$ , $\epsilon_x(y)$ converges to $\hat{y}^j$ on the neighborhood of $\hat{y}^j$ as follows:

$$
\exists\, c<0, \forall\, \|y-\hat{y}^j\|_2<c,\ \epsilon_x(y)=\epsilon_x(y;\hat{y}^j)=y-\hat{y}^j.
\tag{31}
$$

## A.4 VISUALIZATION OF ITERATIVE DENOISING

In this section, we show more visualizations of the iterative denoising process. As shown in Figure 5, some initial noisy actions lead to incorrect denoised action in the 3D Diffuser Actor with the diffusion model. The diffusion-based model picks a blue block when requires red blocks, and an olive block when requires green blocks. The modeling of time-varying complex noise causes the diffusion model to ignore color differences to a certain extent.

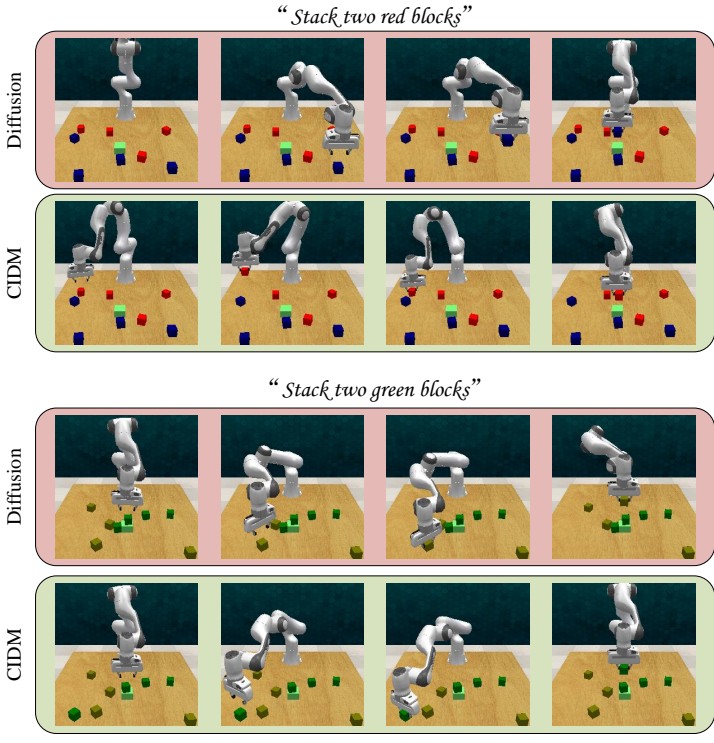

Figure 5: **Visualization of action sequence.** CIDM is less likely to denoise to wrong actions compared with the diffusion-based method.

