# OpenReview forum: "Consistent Iterative Denoising for Robot Manipulation"
_ICLR.cc/2025/Conference — ICLR 2025 Conference Withdrawn Submission_

### Official Review · Reviewer_RTKz · 2024-11-02

**Soundness:** 1
**Presentation:** 1
**Contribution:** 1
**Rating:** 3
**Confidence:** 4

**Summary:**

The authors argue that the denoising targets of conventional diffusion models are inconsistent, making them unsuitable for robotic manipulation tasks, as they 1) vanish near local optima and 2) are time-varying. To address this, the authors propose the Consistent Iterative Denoising Model (CIDM), which learns from a time-invariant denoising field combined with a radial loss function. In this proposed denoising field and radial loss function, distant GT actions have less influence than closer ones. The authors compare CIDM's performance against state-of-the-art text-conditioned visual robotic manipulation methods, such as 3D Diffuser Actor and RVT2, in the RLBench settings used in PerAct and GNFactor.

**Strengths:**

Diffusion models for robotic manipulation indeed behave very differently from denoising in the pixel space of conventional diffusion models for image generation. Unlike pixel-space diffusion, where values are confined within a compact [0,1] range, gripper pose space is unbounded. This often causes diffusion models to exhibit underconvergent behavior as illustrated in Figure 4. The proposed method appears to offer some mitigation for this important issue.

**Weaknesses:**

### **Weakness 1. Lacking Probabilistic Justification**

The authors argue that the score function of conventional diffusion models being zero at local minima is **biased** and should instead always point toward the nearest target. They state:
>
> "The first problem is that the score function $\nabla_{y_t} \log p_{t} (y_t)$ is biased as a denoising field... Since the reasonable denoising field always makes noisy action closer to its target successful
action..."
>
However, this claim is debatable. I would contend that CIDM itself introduces bias while conventional diffusion models are unbiased. For a model to be unbiased, the denoising field should be almost zero near saddle points, as there’s no justification for favoring one specific target. In conventional diffusion models, added noise serves to break such ties. Conversely, CIDM imposes a strong preference toward nearby targets among multiple possible answers, making the output highly sensitive to the initial conditions of the denoising process. This arbitrary choice of denoising field introduces bias in CIDM, unless the distribution of initial points is meticulously selected (as in flow-matching models). Alternatively, one could adopt the Annealed Langevin MCMC viewpoint proposed by Song & Ermon (2019). In this case, however, one should carefully choose the form of noise and denoising target so as to guarantee the learned to model to be unbiased. These considerations are not thoroughly addressed in the paper. Consequently, there's no assurance that the samples $y$, generated by CIDM, follow the actual target policy $y\sim p_{data}(y|x)$.

### **Weakness 2. Claimed Benefit not Well-supported**
As discussed in Weakness 1, CIDM introduces bias. However, as demonstrated by the Cold Diffusion paper, neural networks can still produce reasonable samples across various corruption processes, even if biased. Thus, bias isn’t necessarily detrimental when a meaningful trade-off is achieved. However, for CIDM, the specific benefits of this trade-off remain unclear.

Firstly, it is questionable whether the issue presented in Figure 4 is due to inconsistent training objectives. Rather, it could be due to the inference-time denoising scheduler. For instance, I observe that increasing the number of denoising iterations or lowering the temperature at smaller noise scales often resolves the underconvergence issue shown in Figure 4. Better denoising strategies, such as DDIM, could also be an option.

Secondly, the authors argue that conventional denoising target is difficult to learn, and suggest that CIDM alleviates this issue by using a more consistent target. However, I’m not convinced that inconsistency is the only factor at play here. The primary issue could instead be the precision of the action. Diffusion models often struggle with generating highly precise actions due to their inherently noisy and complicated denoising pipeline. In contrast, models specifically optimized for precision, like RVT2, outperform CIDM and 3D Diffuser Actor in precision tasks such as block stacking as suggested in the experimental result. If the authors argue that inconsistent denoising targets hinder learning, they should provide evidence that biasing the target with a more consistent approach indeed reduces learning variance, i.e., by showing that CIDM demonstrates improved data efficiency or lower performance variance across different seeds.

### **Weakness 3. Insignificant Result**
The experimental results are not significant, as only 25 test episodes were conducted per task. For the 18 tasks in the PerAct setting, this amounts to 450 trials. With CIDM achieving an 82.3% success rate, the 90% confidence interval is 0.78991 ≤ p ≤ 0.85134. Thus, a 1% improvement over state-of-the-art methods like RVT2 and 3D Diffuser Actor does not offer substantial evidence of CIDM’s superiority.

Even if the reported performance gain holds, it does not sufficiently justify the bias introduced by CIDM. For example, if an expert policy selects a red block with 90% probability and a yellow block with 10%, we would expect the learned policy to favor red blocks proportionally. This expectation does not hold for CIDM. Every generative model has precision-diversity trade-off, and the RLBench success rate primarily measures precision over diversity. Therefore, sacrificing sample diversity for only a 1% performance gain does not make a lot of sense for me.

**Questions:**

Which architecture did you use for the denoising network? For a fair comparison, it would be helpful to know how the architectures and the number of parameters are controlled across models.

---

### Official Review · Reviewer_HNF4 · 2024-11-03

**Soundness:** 2
**Presentation:** 3
**Contribution:** 2
**Rating:** 5
**Confidence:** 3

**Summary:**

This paper aims to solve the inconsistent noise supervision issue in diffusion models. The inconsistency comes from two sources. One source is the multi-modal action labels. The other is time-varying noise in denoising steps. They propose a novel consistent iterative denoising model and a new radial loss to address this issue. The proposed method is tested on RL Bench against other baselines.

**Strengths:**

1. The paper clearly illustrates the problem, their motivation to propose the new components to diffusion models and the contributions.
2. The paper provides theoretical analysis to formalize the problem.
3. The paper shows good results on RL Bench and does ablation studies over the different proposed components to show the importance of each part.

**Weaknesses:**

1. In related work, authors list a lot of recent related work in diffusion models. However, some related work is summarized not very clearly. For example, “Inversion by Direct Iteration (Delbracio & Milanfar, 2023) pursues a simpler form to get rid of the limitations of traditional diffusion.” this sentence is confusing because it is not clear to me what things the paper tries to simplify and what limitations they are getting rid of.  Another issue is that the paper mentions that the recent work in diffusion models try to speed up the denoising process and provide in-depth analysis of diffusion models. However, these are not directly related to the inconsistency problem this paper tries to solve. Therefore, I think the paper should reorganize this section so that the connection and difference between related work and the proposed method is more clear.
2. The main advantage of the proposed method as mentioned by the paper is consistent supervision from multiple successful actions (i.e., multi-modality). However, RL Bench demonstrations is not a demonstration dataset that has obvious multi-modality. A recent paper has proposed a dataset benchmark[1] for evaluation of multi-modal behaviors. It would be interesting to see how the proposed methods and the baselines behave in this benchmark
3. The proposed method’s improvement over previous methods on Multi-view is not very significant with 82.3% average success rate compared to RVT2’s 81.4%. For each task, the proposed method has the highest success rate only in 7 out of 16 tasks. Therefore, it seems that the performance improvement is limited.
4. In the results sections, the paper only includes the mean but it is reasonable to also include the std for the success rate as this is usually reported in the other papers.

[1] Xiaogang Jia, Denis Blessing, Xinkai Jiang, Moritz Reuss, Atalay Donat, Rudolf Lioutikov, and Gerhard Neumann. Towards diverse behaviors: A benchmark for imitation learning with human demonstrations. In The Twelfth International Conference on Learning Representations, 2024.

**Questions:**

1. the paper mentions they use CLIP encoder to extract the embedding from text instructions and image observations. However, it doesn’t mention how they process the robot state information in their framework. Moreover, if there are multiple views, how do they fuse the embedding from different views? The paper needs to add some clarification for those details.
2. For qualitative results, the author only shows the stack blocks tasks. It would be interesting to see more qualitative rollouts of other tasks. The paper mentions the method is good for the tasks that has multiple success actions. However, the failure case it show when compared to 3D Diffusor Actor in Appendix A.4 is not multi-modal actions. To solidify the paper’s claim, it is better to include some multi-modal actions example and visualize the denoising process.

---

### Official Review · Reviewer_Z53t · 2024-11-03

**Soundness:** 3
**Presentation:** 3
**Contribution:** 3
**Rating:** 6
**Confidence:** 3

**Summary:**

The paper presents a novel Consistent Iterative Denoising Model (CIDM) aimed at improving action prediction in robot manipulation tasks by addressing issues with diffusion models, specifically noise inconsistency and timestep variations. CIDM introduces two core innovations: (1) a consistent denoising field, which ensures clear denoising directions and temporal consistency across actions, and (2) a radial loss function that emphasizes actions with minimal noise to achieve more accurate iterative denoising.

**Strengths:**

- The paper introduces a novel approach to robot manipulation using a diffusion model, addressing limitations of traditional methods by incorporating a consistent denoising field and a radial loss function.
- Empirical rigor is demonstrated through extensive experiments on the RLBench benchmark, showing clear performance gains over baseline methods. The ablation studies further validate the contribution of each CIDM component, enhancing confidence in the results.
- By addressing practical challenges in action prediction for complex robot tasks, CIDM enhances the applicability of diffusion models.

**Weaknesses:**

- While the paper presents a novel application of iterative denoising to robot manipulation, it lacks a theoretical analysis(Like some other articles on diffusion dynamics$^{[1]}$). Highlighting unique theoretical insights or algorithmic innovations would better justify CIDM’s position in the field.

- The introduction of a radial loss function, while conceptually sound, lacks comprehensive theoretical grounding or references to similar existing loss functions used in other domains. This gap makes it challenging to assess the robustness and scalability of the loss. Providing a more detailed theoretical analysis or justifying it with additional related work on spatial consistency in generative models could clarify its effectiveness.

- The current evaluation focuses on RLBench, but it would significantly benefit from testing in other robotic benchmarks or real-world scenarios to assess generalization capabilities. Evaluating CIDM's performance across tasks with varying levels of action complexity, such as multi-step manipulation in dynamic environments, would enhance the robustness claims.

- Temporal consistency is claimed to improve denoising stability across timesteps, but the scalability of this approach remains uncertain for long-duration tasks. Additional evaluations on tasks requiring extended sequences of actions (beyond 100 timesteps) could illustrate CIDM’s scalability and stability in prolonged scenarios.

[1] Liu, X., Gong, C., & Liu, Q. (2022). Flow straight and fast: Learning to generate and transfer data with rectified flow. arXiv preprint arXiv:2209.03003.

**Questions:**

- Can the author provide more detailed explaination of figure1, I'm not sure I understood  (b) correctly, especially the blue circles in it.

- Equation 12 and 13 seem to have some typos, I think it should be $\exists c>0$.

- The value of denoised field (eq 14) is based on the value of 2-norm between noisy action and successful action. The implicit assumption here is that the 2-norm in the action space is well defined. This assumption is not obvious as the common action space may contains position, angle, linear velocity, angular velocity, torque... The 2-norm between two actions doesn't necessarily make sense.

---

### Official Review · Reviewer_fvRZ · 2024-11-03

**Soundness:** 2
**Presentation:** 1
**Contribution:** 2
**Rating:** 3
**Confidence:** 3

**Summary:**

This paper proposes a novel approach to denoising in diffusion models for robot manipulation tasks. The authors suggest replacing the standard noising/denoising process with a Langevin dynamics denoising field based on a signed distance function (SDF). This field serves as a deterministic gradient for denoising, aiming to improve temporal consistency and convergence to the ground truth action. Additionally, the authors introduce an alternative radial loss function to optimize the denoising network. The method is evaluated on RLBench in simulation.

The paper presents a potentially novel idea by introducing an SDF-based denoising field for diffusion in robot manipulation tasks. However, the clarity of the writing and the consistency of the mathematical formulations require improvement to ensure a thorough understanding of the proposed method. The limited number of trials and the ambiguities surrounding the evaluation metrics raise concerns about the robustness of the results. The authors should address the discrepancies observed in the convergence behavior and provide a more thorough explanation of the method's capabilities and limitations. Addressing the weaknesses and questions identified in this review would significantly strengthen the paper's contribution and impact.

**Strengths:**

The proposal of a deterministic denoising schedule using an SDF is an interesting alternative to traditional diffusion methods. This approach has the potential to enhance temporal consistency and guide the denoising process more directly towards the ground truth action. The ablation studies presented provide evidence supporting the effectiveness of individual components of the proposed method in specific scenarios.

**Weaknesses:**

1.  The applicability of the proposed method appears limited to 2D robotics tasks with end-effector movements, such as tabletop manipulation. The authors do not demonstrate how this approach can be extended to other types of actuations, such as gripper control.
2.  The paper seems to present a potential misunderstanding regarding the capabilities of diffusion models. It is suggested that diffusion models may produce the same noisy action for different successful actions. However, diffusion models are capable of learning multimodal action distributions through the denoising process, even in cases of overlapping Gaussians.
3.  Figure 4 raises questions about the convergence behavior of both the proposed method and the standard diffusion model. In scenarios with multiple successful actions (represented by four red triangles), both methods appear to collapse to a single ground truth action. This behavior contradicts the expectation that these models should be able to learn a multimodal distribution and converge to all valid solutions.
4.  The paper lacks clarity on why the proposed method (CIDM) converges to only one ground truth action in Figure 4, despite demonstrating the ability to learn a bimodal distribution in Figure 2. It remains unclear why the method does not capture the four-modal distribution evident in the task.

There are a number of places in the text where the authors could provide clarification.

### Confusing text

1.  Line 83: The statement "robot manipulation prefers to sample initial actions over the entire action space" is unclear. It is possible the authors intend to convey that the training data covers the entire action space, but the phrasing is ambiguous and requires clarification.
2.  Line 175: The phrase "After eliminating the effects of specific successful action $\hat{y}$" is vague. It is unclear what is meant by "eliminating the effects." Specifying the mathematical operation, such as marginalizing out $\hat{y}$, would improve clarity.
3.  Figure 3 caption, line 337: The caption could improve significantly. It states that the list of 14 tasks tasks are "highly representative."  Highly representative of what?
4.  The paper lacks details on the experimental setup, particularly regarding the number of trials and seeds used for evaluation. The authors state that results are based on four trials per task, but it is unclear how many random seeds were used to ensure the reliability of the results.
5.  The metric "success probability" requires further explanation. If it is calculated based on four trials per task, the possible values should be limited to [0, 25, 50, 75, 100]%. However, Table 2 presents values such as 82.7%, suggesting a different calculation method or a larger number of trials.
6.  Equations 12 and 13 contain an error. The 2-norm $\|y - \hat{y}\|$ cannot be less than a negative number ($c<0$).
7.  Equations 12, 13, and 14 define the denoising field in a way that seems counterintuitive. The denoising field should be $\epsilon_x(y) = \hat{y} - y$ to ensure that a single denoising step, $y + \epsilon_x(y)$, results in the ground truth action $\hat{y}$. The gradient should point towards the ground truth, not away from it.
8.  Line 285: The authors claim to be learning a denoising field independent of $\hat{y}$. However, the training data includes $\hat{y}$, suggesting that the model likely learns $\hat{y}$ implicitly. This statement requires clarification or justification.
9.  Table 1 caption: The caption states that underlined text indicates "suboptimal performance for each column,". Does this mean the second-best performance or some other criterion? Additionally, not every column has an underlined number.

The paper could benefit from additional explanations and clarifications to enhance the reader's understanding of the proposed method.
<!-- The authors could have utilized the extra to address some of the ambiguities and provide more detailed insights. -->

## Minor Typos

1.  Line 15: "CIDM" is used before its introduction in line 144.
2.  Line 77: "noises supervision signals" should be "noise supervision signals."
3.  Line 93: "Additionally, We" should be "Additionally, we."
4.  Equation 1, line 161: If referring to the DDPM scheduler, the term inside the square root should be $(1 - \bar{\alpha_t})$, not $(1 - \bar{\alpha_t^2})$.
5.  Line 28: The statement "Robot manipulation mainly involves two steps, acquiring effective scene representation and predicting correct actions" oversimplifies the complexity of robot manipulation, which also involves elements of execution on hardware and reactive control.

**Questions:**

The authors are encouraged to address the weaknesses identified in this review and provide clarifications on the points raised in the weaknesses section.

---

### Note · Authors · 2024-11-26

I have read and agree with the venue's withdrawal policy on behalf of myself and my co-authors.